# A mammalian pseudogene lncRNA at the interface of inflammation and anti-inflammatory therapeutics

**Nicole A Rapicavoli[1], Kun Qu[1], Jiajing Zhang[1], Megan Mikhail[1], Remi-Martin Laberge[2], Howard Y Chang[1]***

[1]Program in Epithelial Biology, Howard Hughes Medical Institute, Stanford University School of Medicine, Stanford, United States; [2]Buck Institute for Research on Aging, Novato, United States

**Abstract** Pseudogenes are thought to be inactive gene sequences, but recent evidence of extensive pseudogene transcription raised the question of potential function. Here we discover and characterize the sets of mouse lncRNAs induced by inflammatory signaling via TNFα. TNFα regulates hundreds of lncRNAs, including 54 pseudogene lncRNAs, several of which show exquisitely selective expression in response to specific cytokines and microbial components in a NF-κB-dependent manner. Lethe, a pseudogene lncRNA, is selectively induced by proinflammatory cytokines via NF-κB or glucocorticoid receptor agonist, and functions in negative feedback signaling to NF-κB. Lethe interacts with NF-κB subunit RelA to inhibit RelA DNA binding and target gene activation. Lethe level decreases with organismal age, a physiological state associated with increased NF-κB activity. These findings suggest that expression of pseudogenes lncRNAs are actively regulated and constitute functional regulators of inflammatory signaling.

*For correspondence: howchang@stanford.edu

Competing interests: The authors declare that no competing interests exist.

## Introduction

Large scale transcriptome analyses has revealed that three quarters of the human genome may be expressed (*Djebali et al., 2012*), much of it as noncoding RNA (ncRNA). Over the past several years, the literature describing the functions of long noncoding RNA (lncRNA) has exploded with detailed reports demonstrating that lncRNA can regulate neural development (*Feng et al., 2006*; *Bond et al., 2009*; *Rapicavoli et al., 2010*; *Rapicavoli et al., 2011*), differentiation (*Dinger et al., 2008*; *Guttman et al., 2009*; *Loewer et al., 2010*; *Guttman et al., 2011*), epigenetic marks on chromatin (*Rinn et al., 2007*; *Tsai et al., 2010*; *Wang et al., 2011*) and transcription factor signaling (*Willingham et al., 2005*; *Kino et al., 2010*; *Gomez et al., 2013*). In addition, the human genome was found to contain over 11,000 pseudogenes, of which 833 were expressed and associated with active chromatin (*The ENCODE Project Consortium, 2012*). Pseudogenes have traditionally been defined as ancestral copies of protein coding genes that arise from a gene duplication event or by a retrotransposition event that is followed by subsequent accumulation of mutations that render the pseudogene transcriptionally inactive. More recent studies have revealed that many pseudogenes are expressed as lncRNAs and can have a role in gene silencing (*Duret et al., 2006*), and cancer (*Poliseno et al., 2010*; *Kalyana-Sundaram et al., 2012*).

The pro-inflammatory cytokine TNFα acts through the transcription factor NF-κB to play a key role in innate and adaptive immunity, inflammation, apoptosis, and aging. Dysregulation of NF-κB plays an important role in the pathophysiology of inflammatory disease, when proinflammatory cytokines drive NF-κB activation, which in turn drives production of proinflammatory cytokines. The NF-κB family is composed five proteins in mammals: RelA (p65), RelB, c-Rel, p50 (NF-κB1), and p52 (NF-κB2). NF-κB family members form dimers, either as heterodimers or homodimers, which can act to positively or

**eLife digest** The simplest account of gene expression is that DNA is transcribed into messenger RNA, which is then translated into a protein. However, not all genes encode proteins; for some it is the RNA molecule itself that is the end product. Many of these 'non-coding RNAs' are thought to be involved in regulating the expression of other genes, but their exact functions are unknown.

Pseudogenes are genes that have lost their protein-coding abilities as a result of mutations they have accumulated mutations over the course of evolution. They were previously referred to as 'junk DNA' or 'dead genes' because they were thought to be completely non-functional, lacking even the ability to encode RNA. However, recent work has shown that pseudogenes are in fact transcribed into long non-coding RNAs, and these are now the focus of much research.

Here, Rapicavoli et al. report that certain pseudogenes and long non-coding RNAs are involved in regulating the immune response. Specific and distinct pseudogene-derived long RNAs are made when cells are exposed to different kinds of infections. Immune cells such as macrophages and lymphocytes produce a protein called tumor necrosis factor alpha (TNFα), which is involved in triggering fever and inflammation. TNFα exerts these effects by binding to and activating a transcription factor called NF-κB, which then moves to the nucleus and binds to DNA, regulating the expression of genes that encode immune proteins.

Rapicavoli et al. found that the production of a long non-coding RNA called Lethe (after the 'river of forgetfulness' in Greek mythology) increases when TNFα activates NF-κB. Surprisingly, however, Lethe then binds to NF-κB and prevents it from interacting with DNA, thereby reducing the production of various inflammatory proteins.

This is the first time that a pseudogene has been shown to have an active role in regulating signaling pathways involved in inflammation, and raises the possibility that other pseudogenes may also influence distinct feedback loops and signaling networks. It suggests that many novel functions for pseudogenes and long non-coding RNAs remain to be discovered.

negatively regulate target gene expression. Under normal conditions NF-κB dimers are sequestered in the cytoplasm by binding to IκB proteins. Activation by binding of ligand to a receptor on the cell surface leads to a signaling cascade which leads to phosphorylation, rapid ubiquitination, and degradation of the IκB proteins. This reveals a nuclear localization sequence on NF-κB. NF-κB is then translocated to the nucleus where it binds DNA to activate or repress transcription. Importantly, only Rel family members (RelA, RelB, and c-Rel) can activate transcription. p50 and p52 can form heterodimers with RelA family members to activate transcription, or alternatively form homodimers to compete for NF-κB binding sites reviewed in *Hayden and Ghosh (2012)*.

Here, we use paired-end directional sequencing to identify the effects of TNFα stimulation on the entire transcriptome of mouse embryonic fibroblast (MEF) cells. TNFα regulates the transcription of 3596 protein coding genes, 48 annotated lncRNAs, 54 pseudogene lncRNAs, and 64 de novo lncRNAs. We validate a subset of these lncRNAs, and classify them by response to various microbial components and proinflammatory cytokines, dependence on RelA and subcellular localization. We identify an lncRNA pseudogene, Lethe (named after the mythological river of forgetfulness, for its role in negative feedback), which is expressed in response to proinflammatory cytokines TNFα and IL-1β, and the anti-inflammatory agent, dexamthasone, but is not responsive to microbial components, and is primarily found on the chromatin. Lethe is regulated by RelA, independent of pseudogene family members and proximal genes. Additionally, Lethe is dramatically downregulated in aged spleen. Finally, Lethe binds directly to RelA to inhibit NF-κB DNA binding activity. These findings suggest that Lethe may function as a novel negative regulator of NF-κB, to help fine tune the inflammatory response.

## Results

### Identification of lncRNA that are regulated by TNFα

We hypothesized that NF-κB regulates the expression of lncRNAs just as it regulates the expression of coding genes and microRNAs (*Boldin and Baltimore, 2012*). To determine whether NF-κB regulates the expression of lncRNAs, we performed paired-end directional RNA-seq on wildtype (WT) MEFs

before treatment and after treatment for 1.5, 6 and 24 hr with 20 ng/ml of TNFα. On average, more than 20 million reads were mapped to the mouse genome (mm9 assembly) for each treatment condition (*Supplementary file 1A*). First, reads were mapped to the mouse mm9 reference genome using TopHat (*Trapnell et al., 2009*). Using an in-house generated script, RefSeq and Ensemble annotated transcripts' expression in the form of RPKM (reads per kilobase of exon model per million) were obtained and those transcripts with at least a twofold change in expression and an average RPKM > 1, were defined as significant. Reference based de novo transcriptome assembly of mapped reads was performed using two methods. Raw reads were mapped using TopHat and de novo transcriptome assembly of mapped reads was performed using, Cufflinks (*Trapnell et al., 2010*) and Scripture (*Guttman et al., 2010*) in parallel. RefSeq and Ensemble annotated transcripts were downloaded from UCSC table browser, and these annotated transcripts were filtered out from Scripture and Cufflinks-assembled transcriptomes to yield about 1500 novel de novo isoforms that are expressed at an RPKM > 1. Because many isoforms mapped to a single locus, we further filtered the list of novel transcripts by applying promoter regions as defined by H3K4me3 via chromatin immunoprecipitation sequencing (ChIP-Seq), which yielded 184 novel loci. To further refine the candidate transcripts, we extracted the raw reads that mapped to those 184 loci, processed a de novo transcript assembly through Trinity (*Grabherr et al., 2011*) and determined the Coding Potential Calculator (CPC) score of each transcript (*Kong et al., 2007*) to identify 64 novel de novo lncRNAs (*Figure 1A*, *Supplementary file 1B*).

In this way, 3596 protein coding transcripts, 244 ncRNAs and 64 de novo lncRNAs were detected in these experiments. Many RefSeq protein coding genes that had been shown to be regulated by NF-κB were induced by TNFα including *Gadd45b, Sod2, Nfkbia, Relb, Cdkn2a,* and *Il6*. Additionally, the RNA-seq data showed the expected oscillatory gene expression pattern of NF-κB dependent gene expression, and notably the dynamic range by RNA-seq is greater than previously observed with microarrays (*Kawahara et al., 2011*). Interestingly, the 244 RefSeq ncRNAs showed a similar expression pattern as the protein coding genes, where peak expression or repression levels were observed at 1.5 and 24 hr post stimulation. A subset also showed maximal repression at 6 hr in both classes. 84 ncRNAs were at least twofold upregulated when compare to untreated at a least one time point. In contrast, the vast majority (59 out of 64) of de novo lncRNAs were primarily repressed upon TNFα treatment (*Figure 1B–D*, *Supplementary file 1B–D*). A similar result in which most de novo lncRNAs were down regulated after treatment was seen in response to estrogen in breast cancer cells (*Hah et al., 2011*).

Next, we divided the RefSeq ncRNAs by their RefSeq annotation into four classes, pri-miRNAs (40%), RNaseP, SnoRNA, ScaRNA (19%), pseudogene lncRNA (22%) and annotated lncRNAs (19%) (*Figure 1E*). Interestingly, only 11 of 96 pri-miRNAs were upregulated with TNFα treatment. In contrast about 23 of 45 housekeeping RNAs (RNaseP, scaRNA, snoRNA) and 37 of 54 pseudogene lncRNAs were upregulated. Finally, 12 out of 48 annotated lncRNA were upregulated, mirroring what we see in the de novo lncRNAs. To further examine the pseudogene component of the lncRNAs, we created a heatmap of pseudogene lncRNA, and observed the same oscillatory gene expression pattern that was observed in the protein coding genes (*Figure 1F* and *Supplementary file 1E*). We determined that the pseudogene Rps15a-ps4 (herein named Lethe), had the highest expression changes of any pseudogene with an RPKM > 1. Additionally, we observed that *Gapdh* had seven pseudogenes that were identified as induced by TNFα, but we were unable to validate this result with qRT-PCR.

We selected Refseq genes with significant differential expression over the time course (FDR < 0.05, SAMseq) and varied by at least twofold, yielding 3690 significant transcripts (*Supplementary file 1F*). We organized their patterns of temporal expression by mean-centered hierarchical clustering (*Figure 1—figure supplement 1*), and determined which lncRNAs clustered with known NF-κB regulated genes. From our list, we chose to validate and further characterize Cox2 Divergent, Gp96 Convergent, H2-T23/24AS, HoxA11AS, Lethe, Pbrm1 Convergent, Scripture 16,612 and Scripture 60,588.

## LncRNAs distinguish distinct inflammatory stimuli

Our directional paired-end RNA-Seq data revealed TNFα regulation of many lncRNA transcripts which include divergent, antisense, convergent, and intergenic transcripts. Since the functional relationship between genomic organization and expression is unknown, we chose to validate and further characterize the lncRNAs expression alongside the closest protein coding gene under a variety of different stimuli by qRT-PCR. In our subset of lncRNAs, we found that most lncRNAs are co-regulated with their protein-coding gene. This is not surprising since we chose to validate lncRNAs that were close to genes that

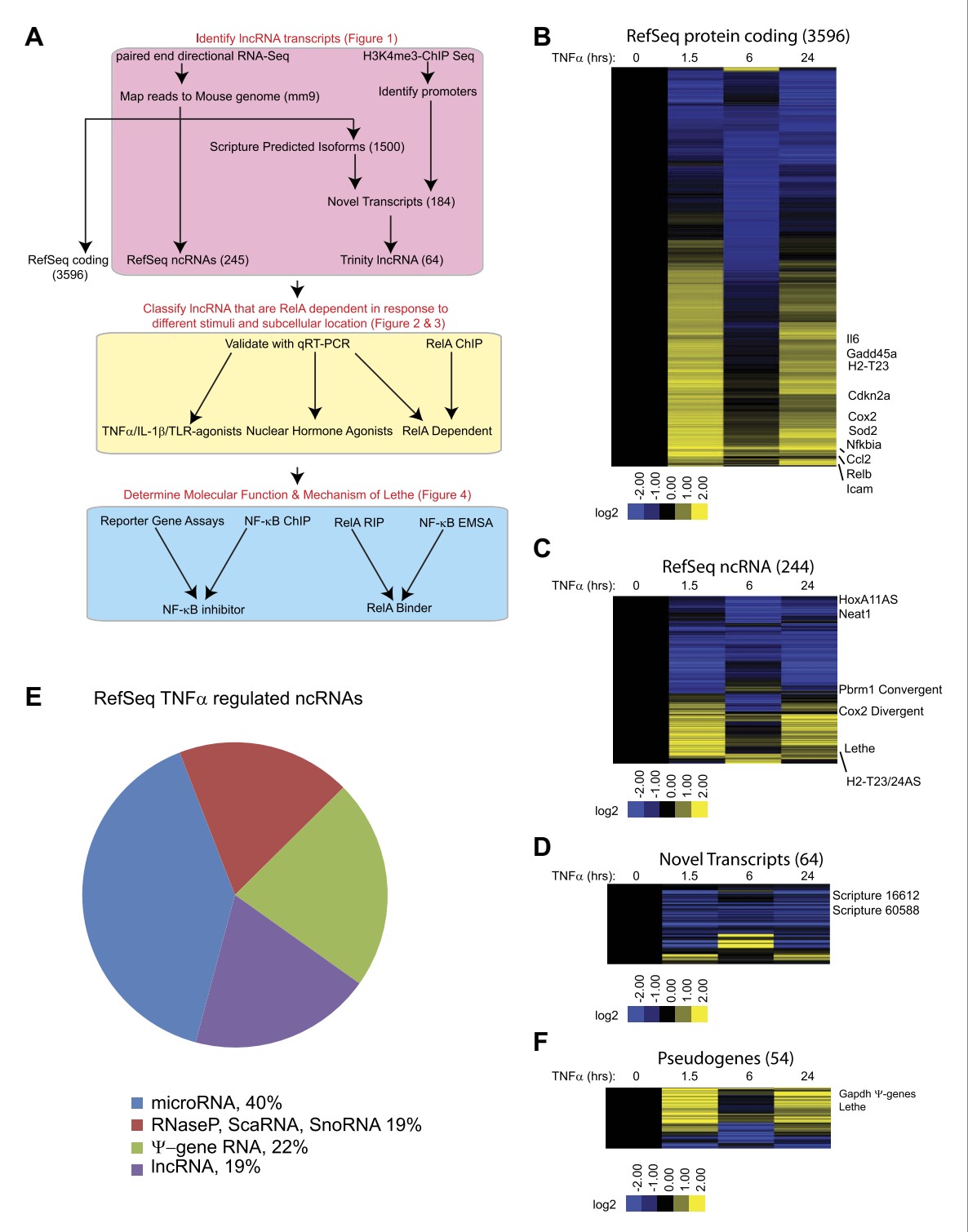

**Figure 1**. TNFα regulates the transcription of many coding and noncoding genes. (**A**) Workflow for strategy for discovery of NF-κB regulated lncRNAs. (**B**) 3596 RefSeq protein coding genes are regulated by TNFα. Values are normalized to the 0 hr time point. (**C**) 244 RefSeq ncRNAs are regulated by TNFα. (**D**) 64 de novo lncRNAs are regulated by TNFα. (**E**) The fraction of all RefSeq ncRNAs for each class of transcript. (**F**) 54 pseudogene lncRNAs are regulated by TNFα.

The following figure supplements are available for figure 1:

**Figure supplement 1**. Heatmap of RefSeq genes.

were regulated by TNFα. One notable exception was Lethe. Although the Lethe is expressed from the same strand as *Gmeb1* and close to the 3′ terminus of *Gmeb1*, qRT-PCR showed that Lethe was specifically induced by TNFα and IL-1β, whereas *Gmeb1* expression did not significantly change, confirming that the Lethe is not an extension of the 3′ UTR of Gmeb1 (*Figure 2A*).

NF-κB signaling can be initiated in response to many different stimuli including in response to proinflammatory cytokines like TNFα and IL-1β as well as in response to microbial components via the TLR family (*Hayden and Ghosh, 2012*). Therefore, we validated our lncRNA candidates in response to TNFα, IL-1β, and agonists of Toll like receptors (TLR) 1, 2, 3, 4, or 7 (*Figure 2A–B*). TLRs are pattern-recognition receptors for pathogen components from bacteria, fungi, or viruses, and play key roles in controlling the innate and adaptive immunity (*Kawai and Akira, 2010*). Indeed, we found that many of the lncRNAs are upregulated in response to distinct stimuli. Most notable, Cox2 Divergent is upregulated in response to proinflammatory cytokines and TLR1-4 agonists. In contrast, Gp96 Convergent is only expressed in response to TNFα. H2-T32/24AS is responsive to only TNFα and TLR3 agonists, whereas HoxA11AS is expressed in response to TLR3 agonists and actually down regulated by TNFα stimulation. Lethe is upregulated in response to the proinflammatory cytokines TNFα and IL-1β, but not TLR agonists, indicating it may have a function in inflammation, but not in native immunity. Pbrm1 Convergent is highly upregulated in response to IL-1β, and to a lesser extent TNFα, and TLR4 and 7 agonists. These results demonstrate that lncRNAs are dynamically and specifically regulated in response to different stimuli, suggesting that the pattern of lncRNAs can serve as an internal representation of a cell's exposure to distinct inflammatory and pathogenic signals.

## LncRNAs are directly regulated by NF-κB

Next, we wanted to determine if the lncRNAs were directly regulated by NF-κB. To address this question, we used two different methods. First, we performed qRT-PCR in *RelA−/−* littermate cells alongside WT cells (*Figure 2—figure supplement 1*). Cox2 Divergent is dramatically upregulated in response to proinflammatory cytokines and TLR1-4 agonists in WT and to an even larger extent, in *RelA−/−* cells indicating that it is not directly regulated by NF-κB component RelA. Additionally, HoxA11AS, H2-T23/24AS, Gp96 Convergent and Scripture 16,612 all show some induction in *RelA−/−* cells. In contrast, induction of Lethe, Pbrm1 Convergent and Scripture 60,588 is largely abrogated in *RelA−/−* cells, indicating that RelA is required to induce these lncRNAs (*Figure 2B*). Second, we performed RelA chromatin immunoprecipitation (ChIP) in WT MEFs. Upon TNFα signaling, RelA was found to bind to the promoters of Nfkbia, Gp96 Convergent and Lethe, but was not detected on the promoters of Cox2 Divergent or Pbrm1 Convergent, or Dll1, a negative control (*Figure 2C*). These results indicate that Lethe and Gp96 Convergent are directly transcriptional targets of RelA, with Lethe being particularly dependent on RelA for induction.

## Lethe lncRNA is largely nuclear and on chromatin

To determine where our candidate genes are located within the cell we performed subcellular fractionation on cells stimulated with TNFα for 6 hr. We found that the subcellular distribution of TNFα−induced lncRNAs vary in a transcript-specific manner. Gapdh was tested as a control and found to be evenly distributed between the nucleus and cytoplasm with little transcript found on the chromatin. Likewise, Cox2 Divergent, Gp96 Convergent, and H2-T23/24AS were evenly distributed between nucleus and cytoplasm. HoxA11AS was mostly nuclear with some transcript found in the cytoplasm, but not on the chromatin. Interestingly, Lethe, Pbrm1 Convergent, Scripture 16,612 and Scripture 60,588 were found mostly on the chromatin, with a smaller fraction in the nucleus. These results indicate that Lethe, Pbrm1, Scripture 16,612 and Scripture 60,588 may be directly involved in gene regulation by interacting with the chromatin (*Figure 2D*). To further determine if it is the nascent transcript or processed transcript that is found on the chromatin, we performed polyA selection on our sub-cellular fractions and analyzed the results by qRT-PCR. Interestingly, polyadenylated Lethe RNA is still preferentially associated with chromatin compared to two control mRNAs, Actin and Nfkbia (*Figure 2—figure supplement 1*), indicating that full length Lethe is associated with chromatin. Finally, we performed H3-RNA immunoprecipitation (RIP) to directly test if lncRNAs are found on the chromatin after cells were stimulated with TNFα for 6 hr. We found that Lethe and Pbrm1 Convergent are both found on the chromatin, while Cox2 Divergent and Gp96 Convergent were not detected, confirming our fractionation results (*Figure 2E*). Results from *Figure 2* are summarized in *Table 1*.

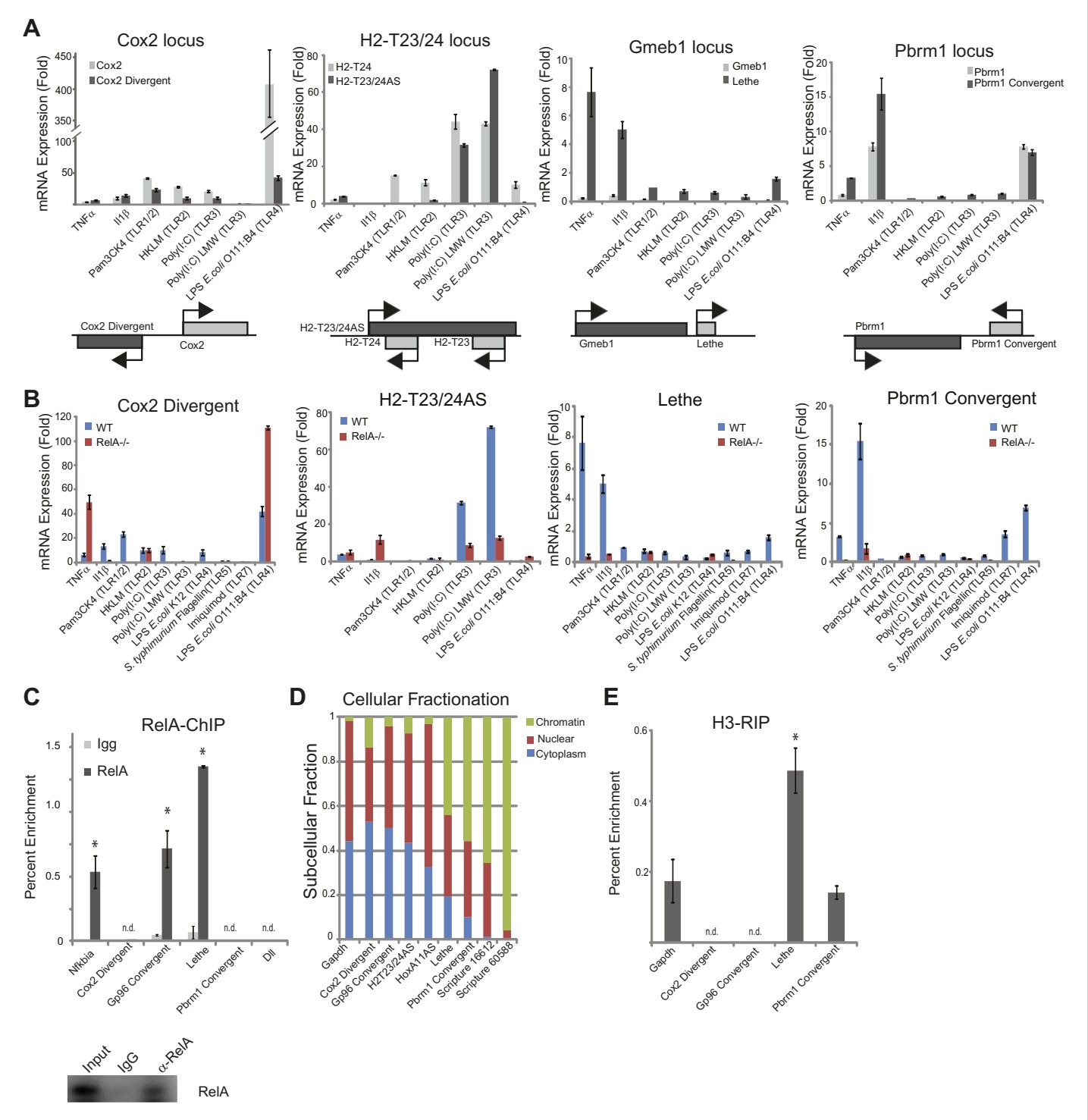

**Figure 2**. LncRNAs distinguish between different stimuli and are regulated by NF-κB. (**A**) Validation of lncRNAs expression alongside the closest protein coding gene under a variety of different stimuli by qRT-PCR. Genomic organization is shown below. MEFs were treated with 20 ng/ml TNFα for 0 and 6 hr. Quantitative Taqman real time RT-PCR of the indicated RNAs is normalized to Actin levels (mean ± SD). (**B**) LncRNAs are regulated by RelA. qRT-PCR in WT and *RelA−/−* littermate cells under a variety of different stimuli. MEFs were treated with 20 ng/ml TNFα for 0 and 6 hr. Quantitative Taqman real time RT-PCR of the indicated RNAs is normalized to Actin levels (mean ± SD). (**C**) Endogenous RelA is recruited to the promoters of lncRNAs. MEFs were treated with 20 ng/ml TNFα for 0 and 15 min. ChIP with α-RelA antibodies was performed and RelA percent enrichment relative to input is shown (mean ± SD, *Nkfbia,* p<0.0518; *Gp96 Convergent*, p<0.007; *Lethe,* p<0.002). (**D**) LncRNAs are found throughout the cell. Cellular fractionation was performed and
*Figure 2. Continued on next page*

*Figure 2. Continued*
fraction found in the chromatin, nucleus and cytoplasm is shown. MEFs were treated with 20 ng/ml TNFα for 6 hr. Quantitative Taqman real time RT-PCR of the indicated RNAs is shown (mean ± SD is shown). (**E**) LncRNAs are found on the chromatin. MEFs were treated with 20 ng/ml TNFα for 6 hr. RNA-IP with α-H3 antibodies was performed. RNA was isolated and Quantitative Taqman real time RT-PCR of the indicated RNAs is shown (mean ± SD, Lethe p<0.004).
The following figure supplements are available for figure 2:
**Figure supplement 1**. (**A**) Western analysis of RelA protein levels. (**B**) PolyA+ Lethe is found on the chromatin.

## Lethe is a pseudogene of Rps15a

Since Lethe is an Rps15a pseudogene, we wanted to determine if there are other pseudogenes that are directly regulated by TNFα. We obtained Taqman probes against non-repetitive sequences unique to each pseudogene member, and examined Rps15a pseudogene family members as well as Rps15a to determine if they are regulated by TNFα (*Figure 3A*, *Figure 3—figure supplement 1*). qRT-PCR analysis showed that *Nfkbia* (a positive control) is dynamically regulated in response to TNFα, as is Lethe. Rps15a and Rps15a-ps6, another Rps15a pseudogene lncRNA, are both transcribed but are not regulated by TNFα (*Figure 3B*).

Lethe is 697 bp long unspliced lncRNA, and its locus on chromosome four lays approximately 500 bp downstream of *Gmeb1* and 8 kb upstream of *Ythdf2* on the minus strand (*Figure 3C*). Lethe is dramatically induced upon TNFα stimulation at 1.5 hr and 24 hr and repressed at 6 hr, in an expression pattern that is characteristic of other NF-κB regulated transcripts. Importantly, expression of its two neighbor mRNA genes was not changed by TNFα stimulation (*Figure 3C*), indicating that Lethe is independently regulated.

## Lethe is induced by the glucocorticoid receptor

It is known that glucocorticoid receptor (GR) and NF-κB share many target gene sites (*Rao et al., 2011*). Therefore we tested whether Lethe could be induced upon stimulation with a number of nuclear hormone agonists including the GR agonist, dexamethasone. We found that Lethe is upregulated in response to anti-inflammatory agent, dexamethasone, but not in response to other nuclear hormone receptor agonists examined, including Vitamin D (Vitamin D Receptor), methyltrienolone (Androgen Receptor) and estradiol (Estrogen Receptor) (*Figure 3D*). Thus, Lethe is a pseudogene lncRNA that is induced by both inflammatory stimuli and an anti-inflammatory therapeutic.

## Lethe is downregulated in aged mice

Recent work has shown that the transcription factor binding motif most strongly associated with aging is NF-κB (*Adler et al., 2007*). To determine if Lethe is expressed in old tissue as a result of constant

**Table 1.** Classification of TNF regulated lncRNAs

|  | TNFα | IL-1β | TLR1 | TLR2 | TLR3 | TLR4 | TLR5 | TLR7 | RelA-dep | H3-RIP |
|---|---|---|---|---|---|---|---|---|---|---|
| Cox2 Divergent | + | + | + | + | + | + | n.d. | n.d. | − | n.d. |
| Gp96 Convergent | + | n.d. | n.d. | n.d. | n.d. | n.d. | n.d. | n.d. | + | n.d. |
| H2-T23/24AS | + | n.d. | n.d. | n.d. | + | n.d. | n.d. | n.d. | − | − |
| HoxA11AS | − | − | − | − | + | − | − | − | − | − |
| Lethe | + | + | − | − | − | − | − | − | + | + |
| Pbrm1 Convergent | + | + | − | − | − | − | − | + | −/+ | + |
| Scripture 16,612 | − | + | + | + | − | − | − | − | n.d. | n.t. |
| Scripture 60,588 | − | − | − | − | + | − | − | − | n.d. | n.t. |

These results demonstrate that lncRNAs are regulated by diverse and specific stimuli. Additionally, lncRNAs are directly regulated by NF-κB. Finallyp, the subcellular distribution of TNFα−induced lncRNAs varies by transcript. n.d., not detectable. n.t., not tested.

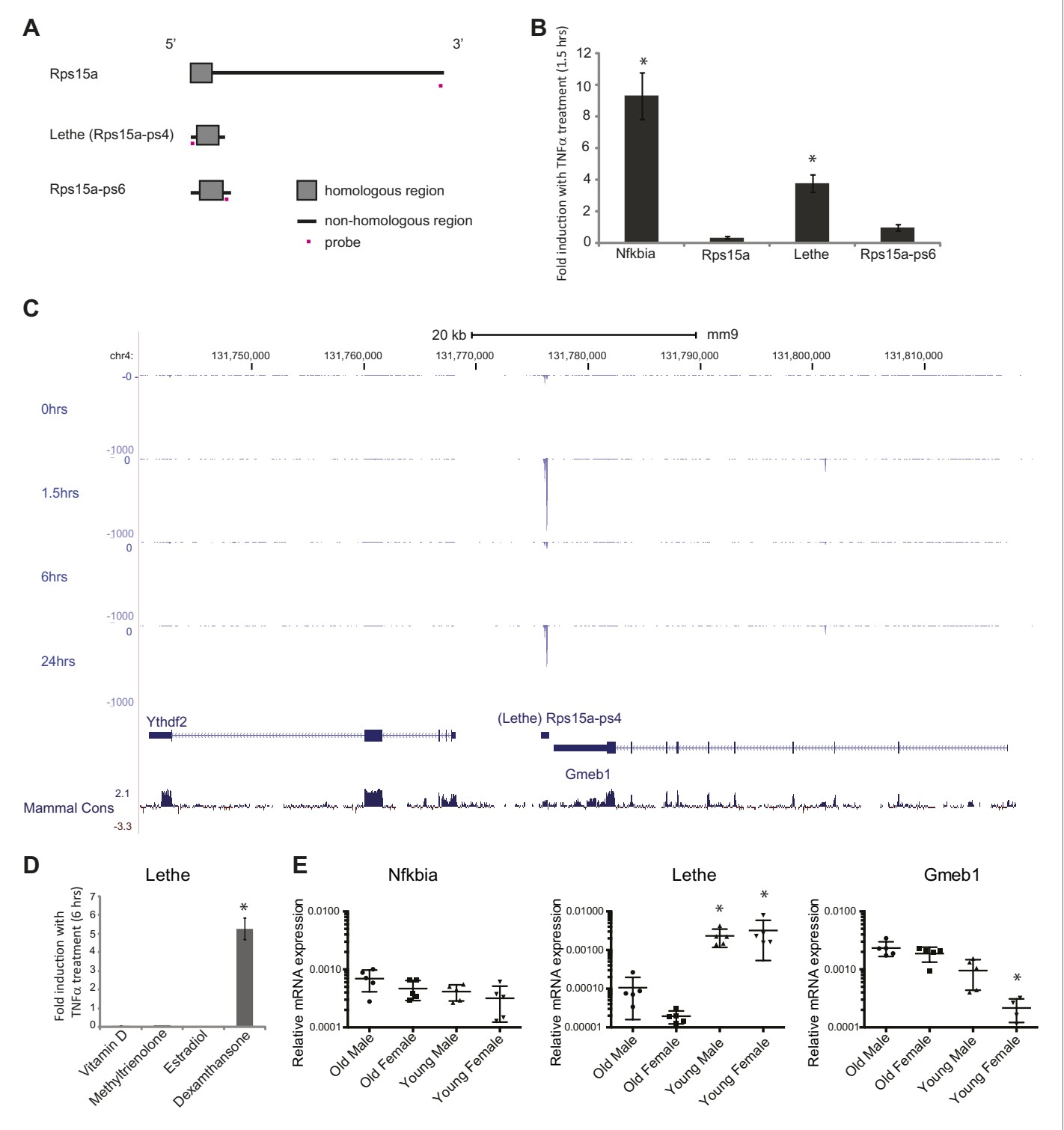

**Figure 3**. Lethe is a pseudogene lncRNAs that is regulated by NF-κB, Glucocorticoid Receptor and in aging. (**A**) Gene structure, homology and Taqman probe design of *Rps15a* and pseudogene family members. (**B**) Lethe is induced by TNFα, but other family members are not. MEFs were treated with 20 ng/ml TNFα for 0 and 1.5 hr. Quantitative Taqman real time RT-PCR of the indicated RNAs is shown normalized to Actin levels (mean ± SD, p<0.012). (**C**) Genomic organization of Lethe with RNA-Seq data at time 0, 1.5, 6 and 24 hr post TNFα treatment. Lethe is located on mouse chromosome 4 between *Gmeb1* and *Ythd2*. *Gmeb1* and *Ythd2* are not induced by TNFα stimulation. (**D**) Lethe is induced by dexamethasone treatment, but not other nuclear hormone receptor agonists. MEFs were treated with either 10 nM vitamin D, 100 nM methyltrienolone, 100 nM estradiol, or 1 µM

*Figure 3. Continued on next page*

*Figure 3. Continued*

dexamethasone for 0 and 6 hr. Quantitative Taqman real time RT-PCR of the indicated RNAs is shown normalized to Actin levels (mean ± SD, p<0.003). (E) Lethe is down-regulated in aged mice. Lethe is expressed in young spleen from male and female mice. Five mice were used for each sex and time point. Quantitative Taqman real time RT-PCR of the indicated RNAs is shown normalized to Actin levels (mean ± SD, Lethe p<0.001, Gmeb1 p<0.003). ANOVA analysis was performed to determine significance.

The following figure supplements are available for figure 3:

**Figure supplement 1**. Alignment of Lethe with Rps15a-ps6.

NF-κB signaling, we tested a panel of tissues, including liver, lung, kidney, skin, spleen, cortex (brain), and skeletal muscle. Lethe is expressed in male and female spleen, but not detectable in other tissues. Interestingly, Lethe is downregulated with age: 20-fold and 160-fold in males and females respectively (*Figure 3E*). Neither *Nfkbia* nor *Gmeb1* expression changes with age or sex.

## Lethe inhibits NF-κB activity

We hypothesized that Lethe acts in trans to regulate NF-κB function. Therefore we performed loss-of-function experiments and measuring expression of canonical NF-κB members. We used chemically modified chimeric antisense oligonucleotides (ASO) which have been shown to be effective at knocking-down expression of nuclear ncRNAs (*Ideue et al., 2009*). ASO blockade inhibited the TNFα induction of Lethe, and we monitored the induction of two NF-κB target genes by qRT-PCR. *Nfkbia* level was significantly higher than TNFα stimulated ASO control in one of the two ASOs tested, while *Nfkb2* was induced significantly for both ASOs tested (*Figure 4A*). This result indicates that Lethe may act as a repressor of NF-κB activity.

Conversely, we overexpressed Lethe or GFP in the presence of an NF-κB luciferase reporter gene after TNFα stimulation. Lethe expression, but not GFP expression, repressed NF-κB reporter gene activity. Additionally, Lethe can increase the repression NF-κB luciferase reporter gene expression in a dose dependent manner. To determine if Lethe's effect on reporter gene activity was specific to NF-κB mediated reporter gene expression, we mutated the κB binding sites out of the reporter plasmid. As expected, the repression was no longer observed, indicating that Lethe requires NF-κB to repress reporter gene expression (*Figure 4B*).

The TNFα inducible repression of NF-κB luciferase reporter gene expression indicates that Lethe may affect the ability of RelA to bind to target promoters. To test this possibility, 293T cells were transfected with Lethe and ChIP was performed with RelA antibodies or IgG. In response to TNFα, Lethe expression significantly decreased RelA occupancy of several NF-κB target genes including *Il6, Sod2, Il8,* and *Nfkbia* (*Figure 4C*). Immunoblot analysis confirmed that that Lethe does not lower RelA protein level (*Figure 4C*). These results indicate that Lethe acts to inhibit NF-κB binding to the chromatin.

## Lethe Binds to RelA and inhibits RelA occupancy of DNA

We reasoned that Lethe may bind RelA directly. RelA-RIP retrieved Lethe, but not Gapdh in MEFs stimulated with TNFα for 6 hr. Interestingly, other lncRNAs, Cox2 Divergent and Gp96 Convergent did not bind to RelA (*Figure 4D*). To further explore the relationship between Lethe and RelA, NF-κB DNA binding was assessed by electro mobility shift assays (EMSA). 293T cells were transfected with either CMV_GFP or CMV_Lethe. 48 hours post transfection cells were stimulated with TNFα for 15 min before nuclear lysates were prepared. As expected, TNF-stimulated extracts contained NF-κB activity, which are shifted by radiolabelled NF-κB probes, specifically competed away by cold NF-κB probes, and supershifted by anti-RelA antibody. Notably Lethe expression blocks DNA binding of the RelA homodimer, but not other isoforms (*Figure 4E*). These results indicate that Lethe may act as an inhibitor of NF-κB by binding directly to the RelA homodimer, and blocking RelA's ability to bind DNA (*Figure 4F*).

## Discussion

Recent large scale RNA-Seq experiments have revealed that lncRNAs are dynamically expressed in normal tissues through development, differentiation, in response to different stimuli and as an organism ages (*Guttman et al., 2009*; *Hah et al., 2011*; *Guttman et al., 2011*; *Chang et al., 2013*).

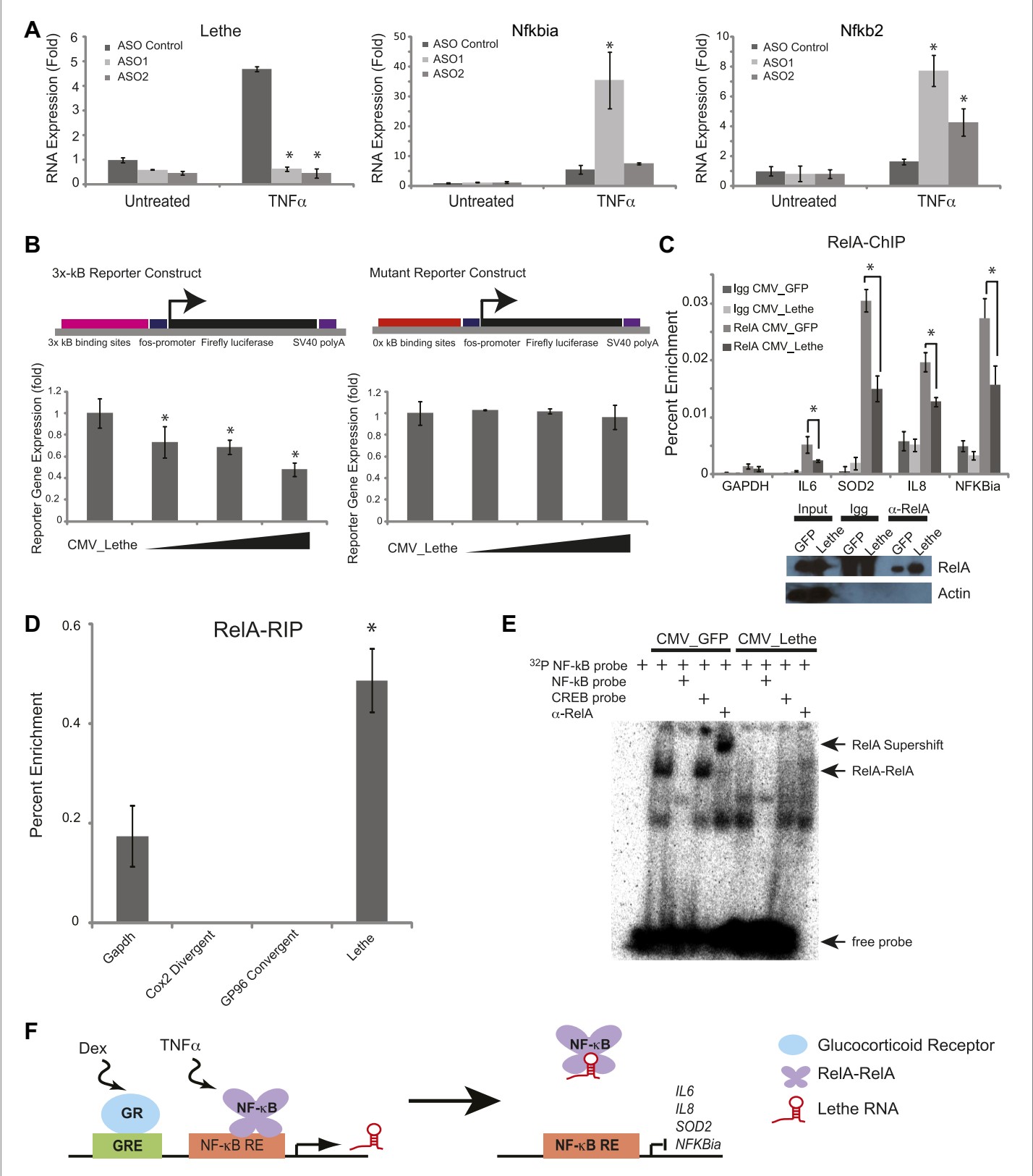

**Figure 4**. Lethe Binds to RelA and inhibits RelA occupancy of DNA. (**A**) Increased expression of NF-κB regulated genes in Lethe knockdown cells. Quantitative Taqman real time RT-PCR of the indicated RNAs is shown normalized to Actin levels (mean ± SD, p<0.05 is shown) (**B**) Lethe inhibits TNFα induced reporter gene expression. RLU of 3x-κB reporter activity and mutant reporter activity (mean ± SD, p<0.05 is shown) in CMV_Lethe transfected
*Figure 4. Continued on next page*

*Figure 4. Continued*

293T cells. Reporter constructs are diagrammed above. (**C**) Endogenous RelA recruitment to the promoters of target genes is reduced in the presence of Lethe. 293T expressing CMV_GFP or CMV_Lethe were treated with 20 ng/ml TNFα for 15 min. ChIP with α-RelA antibodies was performed and RelA percent enrichment relative to input is shown (mean ± SD; *Il6*, p<0.033; *Sod2*, p<0.001; *Il8*, p<0.003; *Nfkbia*, p<0.015). (**D**) Lethe binds to RelA. MEFs were treated with 20 ng/ml TNFα for 6 hr. RNA-IP with α-RelA antibodies was performed. RNA was isolated and Quantitative Taqman real time RT-PCR of the indicated RNAs is shown (mean ± SD, p<0.020). (**E**) Lethe expression blocks DNA binding of the RelA homodimer to its target. NF-κB EMSA of GFP or Lethe transfected 293T nuclear extracts treated with 20 ng/ml TNFα for 15 min. Extracts were pretreated with unlabeled NF-κB (specific) or CREB (nonspecific competitor), or α-RelA antibodies for 15 min prior to incubation with probe. (**F**) Model for Lethe regulation of gene expression. Upon addition of TNFα or dexamthasone, Lethe is transcribed. Lethe can then bind to RelA–RelA homodimers and block binding to other NF-κB response elements, inhibiting NF-κB.

However, most large scale sequencing experiments discard the pseudogene transcriptional contribution. In depth analyses have revealed that pseudogenes are important drivers and suppressors of human cancers (*Poliseno et al., 2010*; *Tay et al., 2011*; *Kalyana-Sundaram et al., 2012*). In this study we performed paired-end directional sequencing to identify novel transcripts that are regulated by TNFα signaling. We identified 48 annotated lncRNAs, 64 de novo lncRNAs and 54 pseudogene lncRNAs that are differentially regulated with TNFαstimulation. While prior studies have reported lncRNA induction by endotoxin (*Guttman et al., 2009*; *De Santa et al., 2010*), the specificity of the response and requirement of NF-κB were not known. Here, we have characterized a number of transcripts that are induced by a panel of microbial components and inflammatory cytokines. Notably, many lncRNAs are uniquely regulated by specific stimuli in a RelA-dependent fashion. Hence, the repertoire of specific transcriptional programs downstream of inflammatory and innate immunity signaling is expanded by the recognition of lncRNAs reported here.

Lethe is a pseudogene lncRNA that comprises an unexpected a regulator of the inflammatory response. Lethe is uniquely induced by the inflammatory cytokines TNF-α and IL-1β, and Lethe inhibits NF-κB by physical interaction to inhibit RelA binding to DNA. Lethe is thus a negative feedback inhibitor of NF-κB signaling, and its mode of action is that of a decoy lncRNA (*Wang and Chang, 2011*). Lethe's mechanism of action is reminiscent of similar to Gas5 or PANDA lncRNAs that titrate glucocorticoid receptor or NF-YA transcription factors away from their cognate binding sites, respectively (*Kino et al., 2010*; *Hung et al., 2011*). Endogenous polyadenylated Lethe RNA is highly associated with the chromatin fractions. Nonetheless, the current data do not distinguish whether the functional pool of Lethe is on chromatin, in nucleoplasm, or both. When Lethe is fused to a SV40 polyadenylation signal (which will efficiently cause primary transcript processing and polyadenylation), Lethe expression inhibited NF-κB -dependent gene expression in a dose-dependent manner (*Figure 4B*), suggesting that chromatin tethering as a primary transcript is not strictly required.

Aging is associated with and requires activation of NF-κB-mediated gene expression programs (*Adler et al., 2007*; *Southworth et al., 2009*); the age-associated loss of Lethe expression we observed may be one of the causes for increased NF-κB activity in aging. Intriguingly, Lethe is also selectively induced by dexamethasone, an anti-inflammatory glucocorticoid agonist, but not by other nuclear hormone receptor agonists tested. These results raise the tantalizing concept that an anti-inflammatory therapeutic acts in part by directly activating the negative feedback system of pro-inflammatory signaling (*Figure 4F*). Lethe's age dependent down regulation is especially interesting because inflammatory diseases such as lupus, rheumatoid arthritis, and ulcerative colitis have a higher incidence in females indicating that Lethe may have a protector role in the inflammation response that is lost with age. LncRNAs may be particularly suited to play such a balancing role in cellular signaling because its regulatory elements can receive and potentially integrate multiple input signals. The fact that lncRNA expression tends to be more tissue- and state-specific than mRNAs suggests lncRNAs are well positioned to adjudicate and diversify signaling networks in a context-specific manner (*Ravasi et al., 2006*; *Cabili et al., 2011*; *Djebali et al., 2012*).

Our work adds to the concept that some pseudogenes may have function as lncRNAs.

Current literature suggests that pseudogenes are under very little selective pressure and therefore can rapidly evolve. While many pseudogenes are likely to be genetic fossils that do not have any function, perhaps the best example of pseudogene functionalization as lncRNA comes from Xist. Xist evolved from the pseudogene degeneration of a protein coding gene in the placental mammalian lineage (*Duret et al., 2006*), and is now essential for dosage-compensation and X chromosome inactivation in

female mammals. Moreover, some pseudogenes may act as endogenous inhibitors of microRNAs in vivo (*Poliseno et al., 2010*; *Salmena et al., 2011*; *Ebert and Sharp, 2010*). However, there is often no correlation between pseudogene and cognate wild-type genes across many tissues (*Kalyana-Sundaram et al., 2012*), suggesting that the primary role of pseudogenes is not to act as an endogenous inhibitor of microRNAs. The specificity of Lethe expression highlights the need to accurately annotate pseudogenes in high throughput analysis and the need to further explore the roles of pseudogenes found throughout the genome.

# Materials and methods

## Animals and cell lines

All animal experiments were approved by the Stanford University Institutional Animal Care and Use Committee and the University of Michigan. Young and old mouse strain and husbandry conditions as described in (*Miller et al., 2011*). Young and old mice were 4-month and 22-month-old respectively. Primary WT and RelA−/− MEFs were harvested from littermate 13.5-day-old embryos using standard methods and propagated in DMEM (Invitrogen, Carlsbad, CA) plus 10% FBS. MEFs were passaged a total of four times before all experiments. All experiments were performed in a minimum of two independently derived MEF lines of the same genotype. 293T cells were grown in DMEM plus 10% FBS. Cells were transfected with Fugene6 (Promega, E2691) per the manufacturer's instructions, and were harvested two days post-transfection.

## Reagents

TNFα (210-TA-050) and IL-1β (201-LB-005) were ordered from R&D Systems, LPS from *Escherichia coli* 0111:B4 (L5293) was ordered from Sigma-Aldrich, and the Human TLR1-9 Agonist kit (tlrl-kit1hw) was purchased from InvivoGen. Dexamethasone was obtained from Sigma (D4902). vitamin D, methyl-trienolone, and estradiol were a gift from Brian Feldman. Antibodies specific for RelA (p65) (ab7970; Abcam), Histone H3 (ab1791; Abcam), Histone H3 (tri methyl K4) antibody (ab8580; Abcam), Rabbit Control IgG (ab46540; Abcam), and Actin (A5316; Sigma) are from the indicated sources.

3xκB Luciferase reporter, pTK-Renilla, and pCMV_GFP were obtained from lab stocks, were sequenced and compared to NCBI for confirmation. pCMV_Lethe was cloned from genomic DNA using primers listed in *Supplementary file 1G*.

## RNA-sequencing and analysis

Total RNA was isolated from MEFs treated with 20 ng/ml TNFαfor the indicated times before RNA extraction with Trizol (10,296-010; Invitrogen), followed by RNAeasy kit (74,104; Qiagen) and treated with Turbo DNAse Free Kit (AM1907; Ambion). Poly-A RNA was selected for using the MicroPoly(A) Purist kit (AM1919; Ambion). 200 ng polyA RNA was used for each library. Paired End Directional library construction was performed for dUTP libraries as described (*Levin et al., 2010*) except libraries were size selected by gel purification after ligation and after PCR amplification. Libraries were sequenced with an Illumina Genome Analyzer II by the Stanford Functional Genomics Facility.

Sequencing reads were mapped to the mouse genome (mm9 assembly) using TopHat (version 1.1.3) (*Trapnell et al., 2009*). Each sample generated 23–32 million mapped sequences. Reference-based de novo transcritome assembly was performed using Cufflinks (version 0.9.3) (*Trapnell et al., 2010*) and Scripture (*Guttman et al., 2010*). RefSeq and Ensemble annotated transcripts were filtered out from Scripture and Cufflinks assembled transcriptomes. Transcripts with less than RPKM > 1were also removed. De novo transcript assembly was processed through Trinity (*Grabherr et al., 2011*) and CPC scores were determined (*Kong et al., 2007*). To determine the number of statistically significant differentially expressed genes for hierarchical clustering, we performed SAMseq, a nonparametric method for estimating significance in RNA-seq data (*Li and Tibshirani, 2011*) and discovered 3690 significant transcripts with FDR < 0.05.

## ChIP-seq

DNA was cross-linked for 10 min with 1% formaldehyde and stopped in 0.125 M glycine. Purified chromatin was sonicated to ~250 bp using the Bioruptor (Diagenode, Inc., Delville, NJ) and incubated with the IgG or Histone H3 (tri methyl K4) as previously described in http://farnham.genomecenter.ucdavis.edu/pdf/FarnhamLabChIP%20Protocol.pdf. ChIP-seq libraries were made and sequenced as above after second strand synthesis. Size-selected libraries of 200–300 bp length were used for Illumina deep-sequencing.

Raw reads from ChIP-Seq were mapped to mouse genome (mm9 assembly) using Bowtie (version 0.12.6) (*Langmead et al., 2009*) and H3K4me3 peaks were called out using MACS (*Zhang et al., 2008*).

## Real-time quantitative RT-PCR

MEFs were treated with TNF-α (20 ng/ml), 10 ng/ml human IL-1β, 100 ng/ml LPS (*E. coli* 055:B5), 100 ng/ml Pam3CSK4, $10^8$ cells/ml HMLK, 25 µg/ml poly(I:C), 25 µg/ml poly LMW (I:C), 10 µg/ml LPS (from *E. coli* strain K12), 100 ng/ml recombinant flagellin (*Salmonella Typhimurium*), 5 µg/ml imiquimod-R837, 10 nM vitamin D, 100 nM methyltrienolone, 100 nM estradiol, or 1 µM dexamethasone for 6 hr. Total RNA was prepared as described above. RNA was analyzed on a LightCycler 480 by RT-qPCR using total RNA (100 ng), Taqman One Step RT-PCR master mix (4309169; Life Technologies). Assays are listed in *Supplementary file 1G*. Reactions were in triplicate for each sample and performed a minimum of three times. Data were normalized to Actin levels.

Young and old mice tissue was mixed with QIAzol (Qiagen) in a 2-ml tube containing a 5-mm stainless steel bead (Qiagen) and was then disrupted on a tissue lyser. CHCl3 was mixed to the homogenate and after centrifugation the aqueous solution was apply to a RNeasy column (Qiagen). The RNA purification was then finished on an automated QIAcube system (Qiagen) and included a DNAse treatment.

## Chromatin immunoprecipitation (ChIP-qPCR)

MEFs were treated with TNF-α (20 ng/ml) for 15 min. ChIP was performed as above for ChIP-Seq. Chromatin was sonicated to 500 bp.

293T cells were treated with TNF-α (20 ng/ml) for 10 min. DNA was cross-linked for 10 min with 1% formaldehyde and stopped in 0.125 M glycine. Purified chromatin was sonicated to ~500 bp using the Bioruptor (Diagenode, Inc) and incubated with 2 µg RelA antibodies or IgG at 4°C overnight. Immunoprecipitation was performed with the Rainin Purespeed tips (PT-2-A5) per manufacturer's instructions.

DNA was analyzed on a LightCycler 480 (Roche) using LightCycler 480 SYBR Green I Master Mix (4707516001; Roche) per manufacturer's instructions. Primers are listed in *Supplementary file 1G*.

## Subcellular fractionation

MEFs were treated with TNF-α (20 ng/ml) for 6 hr. Cells were fractioned as previously described (*Méndez and Stillman, 2000*). RNA was extracted and analyzed as above.

## RNA immunoprecipitation (RIP-RT-qPCR)

MEFs were treated with TNF-α (20 ng/ml) for 6 hr. IP was performed as described above for ChIP except all buffers were pH 7.0 and cells were cross-linked with 1% glutaraldehyde. RNA was extracted and analyzed as above.

## Knockdown experiments

MEF cells were nucleofected using the Nucleofector for mouse embryonic fibroblasts per manufacturer's instructions (VPL-1004, Lonza) except 500,000 cells were nucleofected per condition in 1 µM ASO and plated in one well of a 6-well plate. The ASOs (IDT) were designed as described in (*Ideue et al., 2009*) to increase stability. 48 hr post nucleofection, cells were treated with 20 ng/ml TNF-α for 6 hr prior to RNA extraction.

## Reporter assays

293T cells were transfected with Fugene6 per the manufacturer's instructions 24 hr post plating with 1 µg of 3xκB Luciferase reporter construct, 50 ng of pTK-Renilla and a total of 150 ng of expression plasmids for GFP and Lethe in a 12-well plate. Each condition was performed in triplicate. After 48 hr, cells treated with 20 ng/ml TNF-α for 6 hr. Cells were harvested 2 days post-transfection and luciferase was measured per manufacturer's instructions with the Dual-Luciferase Reporter System (E1910; Promega). Luciferase values were normalized to Renilla to control for transfection efficiency. Experiments were repeated three independent times.

## EMSA

293T cells were transfected with Fugene6 per the manufacturer's instructions 24 hr post plating with 10 µg pCMV GFP or pCMV Lethe. After 48 hr, cells treated with 20 ng/ml TNF-α for 10 min, washed

two times with PBS and lysed. Nuclear lysates were prepared as previously described (*Schmitt et al., 2011*). NF-κB DNA binding activity was measured using the Gel Shift Assay System (E3300; Promega) per manufacturer's instructions. Briefly, 8 µg of nuclear lysate were incubated with 10 pg of $^{32}$P-labeled DNA probe for 15 min. For supershift, lysate was preincubated with 1 µg RelA antibody for 10 min. For competitive and non-competitive experiments, 100-fold molar excess unlabeled NF-κB or CREB probe were preincubated with lysate for 10 min. Complexes were separated by electrophoresis on 6% TBE gels (EC6265; Invitrogen) and assayed by PhosophorImager analysis. Assays were repeated three independent times.

## Accession numbers

Deep sequencing data in this study are available for download from Gene Expression Omnibus (http://www.ncbi.nlm.nih.gov/geo) (accession ID: GSE47494).

## Acknowledgements

We thank Brian Feldman of Stanford University and Richard Miller of the University of Michigan for providing reagents. Lisa Zaba, Lingjie Li, Meihong Lin, Kevin Wang, Orly Wapinski, Grace Zheng, and Adam Schmitt provided technical help. We also thank Judith Campisi, Brian Feldman, and Brian Zarnegar for helpful discussions and comments on the manuscript.

## Additional information

### Funding

| Funder | Grant reference number | Author |
| --- | --- | --- |
| Ellison Medical Foundation | | Howard Y Chang |
| Glenn Foundation | | Howard Y Chang |
| Howard Hughes Medical Institute | | Howard Y Chang |
| National Institutes of Health | T32 AG00026 | Nicole A Rapicavoli |
| National Institutes of Health | P01 AG017242, P01 AG041122 | Remi-Martin Laberge |
| Buck Institute | Research on Aging Fellowship | Remi-Martin Laberge |

The funders had no role in study design, data collection and interpretation, or the decision to submit the work for publication.

### Author contributions

NAR, Conception and design, Acquisition of data, Analysis and interpretation of data, Drafting or revising the article; KQ, JZ, Conception and design, Analysis and interpretation of data, Drafting or revising the article; MM, Acquisition of data, Drafting or revising the article; R-ML, Drafting or revising the article, Contributed unpublished essential data or reagents; HYC, Conception and design, Drafting or revising the article

### Ethics

Animal experimentation: This study was performed in strict accordance with the recommendations in the Guide for the Care and Use of Laboratory Animals of the National Institutes of Health. All of the animals were handled according to approved institutional animal care and use committee (IACUC) protocols (#14046) of Stanford University.

## Additional files

### Supplementary files

• Supplementary file 1. (**A**) Sequencing depth for all conditions. (**B**) List of de novo transcripts: Trinity generated transcripts with the genomics coordinates, the number of mapped reads, RPKMs, CPC score and if there is an H3K4me3 peak from ChIP-Seq experiments. (**C**) List of RefSeq coding genes: RefSeq genes with the number of mapped reads, RPKMs and fraction nucleotide coverage for each.

(D) List of RefSeq ncRNA: RefSeq ncRNA with the number of mapped reads, RPKMs and nucleotide coverage for each. (E) List of pseudogenes: pseudogene ncRNA with the number of mapped reads, RPKMs and nucleotide coverage for each. (F) List of significantly expressed transcripts from RefSeq: RefSeq transcripts with the log2RPKM values. (G) List of all primers used: primer sequences used for ChIP, Taqman assays, antisense oligos, and primers for cloning.

### Major dataset

The following dataset was generated:

| Author(s) | Year | Dataset title | Dataset ID and/or URL | Database, license, and accessibility information |
| --- | --- | --- | --- | --- |
| Rapicavoli NA, Qu K, Zhang J, Chang HY | 2013 | A pseudogene lncRNA at the interface of inflammation and anti-inflammatory therapeutics | GSE47494; http://www.ncbi.nlm.nih.gov/geo/query/acc.cgi?token=lfulxmamcamqmpk&acc=GSE47494 | Publicly available at GEO (http://www.ncbi.nlm.nih.gov/geo/). |

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
