## [Decision Letter]

Thank you for sending your work entitled “A pseudogene lncRNA at the interface of inflammation and anti-inflammatory therapeutics” for consideration at *eLife*. Your article has been favorably evaluated by a Senior editor, Detlef Weigel, and 3 reviewers, one of whom is a member of our Board of Reviewing Editors.

The Reviewing editor and the other reviewers discussed their comments before we reached this decision, and the Reviewing editor has assembled the following comments to help you prepare a revised submission.

The manuscript is timely in its subject matter and clearly written: a better understanding of the biological roles for lncRNAs is clearly critical, since only a small fraction of ncRNAs in general have had their biological roles identified.

While several of the authors’ observations are well supported by the data presented, there are several issues that should be addressed in order to assist the reader to better understand some of the conclusions reached.

1) RNAseq analyses:

a. It is surprising that ∼20 million reads per time point are sufficiently deep to accurately detect relatively small changes in expression, when many of these changes are occurring at very low expression levels. It would be useful to have a table that provides the number of reads and RPKMs that are associated with the four classes of RefSeq RNAs cited in the first section of the manuscript (pri-miRNAs, RNAse P etc., pseudogenes and lncRNAs).

b. It would also be helpful to readers to understand what range of nucleotide coverages were being obtained for each annotated member whose expression level changes were being cited as significant (e.g., 90% of annotation or 10% of the annotation).

c. Why were twofold changes used as a threshold for analyses performed on transcripts or gene elements? Since two biological replicates were used, why not use some statistic like IDR to have a principled way to select thresholds and to filter data sets for inconsistent RPKM values? Finally, why set a threshold of lower expression at RPKM>1?

2) It is quite striking that the profile of the 6 hr time point is significantly different from the other two time points (Figure 1). It would be helpful if the authors were to comment on possible reasons why all classes of ncRNAs reported (annotated and novel) react in this fashion, and whether this difference the decreases from time 0 and increase to 24 hr time points were gradual. From these data it appears as if there is a global resetting of the expression state form time point 0 to 6 hr. Is there any evidence for changes from genome-wide epigenetic profiles that can be correlated with the expression results at the genomic locations cited?

3) The localization of Lethe and Pbrm1 to chromatin is very interesting. However, the use of RT-PCR (Figure 2) as an evaluation of the RIP results is potentially problematic since only relatively short stretches of the mRNAs are assayed in this readout. Thus, there are two potential issues using this approach. First, unless the full length of the Lethe 697 nt transcript was targeted by RT-PCR, then it might well be that it was a sub-portion of this region or another transcript that includes the detected Lethe RNA that was detected. Second, depending upon the turnover rate of a primary transcript it would not be unusual to see evidence of the primary Lethe transcript present in the chromatin enriched fraction. The absence of Cox2 Divergent and GP96 Convergent in the chromatin localization is again interesting, but this does not address the question of whether a newly synthesized primary Lethe transcript is being detected and the functioning portion of the Lethe RNA is the smaller signal detected in the nucleus. If the authors have additional evidence that it is the processed version of Lethe that is associated with chromatin, then it would be useful to indicate this.

4) The conclusion that Lethe is a “unique” pseudogene lncRNA that is induced by both inflammatory and anti-inflammatory therapeutic is an over-interpretation of the results presented.

---

## [Author Response]

*1) RNAseq analyses*:

*a. It is surprising that ∼20 million reads per time point are sufficiently deep to accurately detect relatively small changes in expression, when many of these changes are occurring at very low expression levels. It would be useful to have a table that provides the number of reads and RPKMs that are associated with the four classes of RefSeq RNAs cited in the first section of the manuscript (pri-miRNAs, RNAse P etc., pseudogenes and lncRNAs)*.

We thank the reviewers for their suggestion and we have (1) made a table of sequencing depth of each sample as Supplementary file 1A; (2) made tables in [Supplementary-material SD1-data] that provide the number of mapped reads, RPKMs, and nucleotide coverage for the four classes of genes showing in Figure 1. We agree that the depth of RNA-seq can potentially limit the sensitivity of detecting differential gene expression; hence we carried out extensive qRT-PCR analyses for the transcripts studied in this manuscript.

*b. It would also be helpful to readers to understand what range of nucleotide coverages were being obtained for each annotated member whose expression level changes were being cited as significant (e.g., 90% of annotation or 10% of the annotation)*.

As the reviewers suggested, we have added a column to the tables in [Supplementary-material SD1-data], which indicate the degree of coverage. The degree of coverage of each gene was calculated by the number of bases that were covered by 1 or more reads in any of the four samples divided by the total length of that gene. Over 97.3% of all the significant genes showing in Figure 1 have degree coverage > 0.5, which indicates a majority of our significant genes have reasonable coverage by mapped reads. Readers can now also filter our data for their own desired coverage threshold.

*c. Why were twofold changes used as a threshold for analyses performed on transcripts or gene elements? Since two biological replicates were used, why not use some statistic like IDR to have a principled way to select thresholds and to filter data sets for inconsistent RPKM values? Finally, why set a threshold of lower expression at RPKM>1*?

We used two-fold cut off as a simple heuristic to estimate the number of differentially expressed genes. We agree that a principled method with statistical backing is preferable. Hence, we performed SAMseq, a nonparametric method for estimating significance in RNA-seq data (29) and discovered 3690 significant transcripts with FDR < 0.05. The heatmap of these significant transcripts is shown in Figure 1—figure supplement 1. We now report the log2 RPKM data in [Supplementary-material SD1-data], allowing readers to readily download the statistically significant set of transcripts. Transcripts with RPKM = 1 approximately equals to one copy per-cell, and therefore has been widely used as a cutoff.

*2) It is quite striking that the profile of the 6 hr time point is significantly different from the other two time points (Figure 1). It would be helpful if the authors were to comment on possible reasons why all classes of ncRNAs reported (annotated and novel) react in this fashion, and whether this difference the decreases from time 0 and increase to 24 hr time points were gradual. From these data it appears as if there is a global resetting of the expression state form time point 0 to 6 hr. Is there any evidence for changes from genome-wide epigenetic profiles that can be correlated with the expression results at the genomic locations cited*?

TNFα is known to induce NF-κB activity in an oscillatory fashion. This is due to the fact that NF-κB induces the transcription of its own inhibitor IkB, which titrates NF-κB out of the nucleus until further signaling induces IkB degradation, allowing NF-κB to bind target genes again (Hoffmann, Levchenko et al. 2002). Between 1.5 and 6 hrs, one can often catch the difference between a peak and a trough, although the actual periodicity is shorter. We have previously observed this oscillatory gene expression pattern by microarrays [Figure 6 in (Kawahara, Michishita et al. 2009); Figure 4 of (23)]. Thus RNA-seq showed an expected pattern of NF-κB dependent gene expression and notably the dynamic range by RNA-seq is greater than previously observed with microarrays. We have previously shown oscillatory patterns of RelA and Sirt6 occupancy at NF-κB target genes (23). We apologize that this point was not clear and we have revised the text to explain the change at the 6 hr time point.

*3) The localization of Lethe and Pbrm1 to chromatin is very interesting. However, the use of RT-PCR (Figure 2) as an evaluation of the RIP results is potentially problematic since only relatively short stretches of the mRNAs are assayed in this readout. Thus, there are two potential issues using this approach. First, unless the full length of the Lethe 697 nt transcript was targeted by RT-PCR, then it might well be that it was a sub-portion of this region or another transcript that includes the detected Lethe RNA that was detected*.

We agree with the caveat that this experiment only provides evidence that the specific tested region of Lethe is interacting with chromatin, and we have amended the text accordingly. To our knowledge, RNA immunoprecipitation is routinely read out by qRT-PCR (e.g., (20); (Kretz, Siprashvili et al. 2013)). Many investigators shear or digest the RNA to a small fragment in order to map the site of interaction between protein and RNA. The detection of Lethe poses additional challenges because Lethe is a pseudogene transcript. Only small portions of Lethe sequence uniquely distinguish Lethe from the Rps15a and pseudogene family members (Figure 3). We tested many primer sets to distinguish Lethe from family members, and also distinguish RNA from contaminating DNA. The primers we chose were the ones that performed the best. The alternative to RT-PCR, such as Northern blot analysis or tiling array analysis hybridization, would have far worse cross hybridization problems than targeted, gene-specific qRT-PCR.

*Second, depending upon the turnover rate of a primary transcript it would not be unusual to see evidence of the primary Lethe transcript present in the chromatin enriched fraction. The absence of Cox2 Divergent and GP96 Convergent in the chromatin localization is again interesting, but this does not address the question of whether a newly synthesized primary Lethe transcript is being detected and the functioning portion of the Lethe RNA is the smaller, signal detected in the nucleus. If the authors have additional evidence that it is the processed version of Lethe that is associated with chromatin, then it would be useful to indicate this*.

We agree that nascent primary transcripts are tethered to chromatin, and hence the chromatin enrichment signal of each RNA should be compared against other transcripts for relative enrichment. As the reviewers observed, Lethe showed 5 to 10 fold higher chromatin association in cellular fractionation and histone H3 RIP than five other lncRNAs tested at the same time (Figure 2). Two lncRNAs, Cox2 Divergent and GP96 Convergent, showed undetectable signal on H3 RIP; these transcripts were localized to the cytoplasm instead. The chromatin association signal for Lethe is also significantly greater than that of GAPDH (Figure 2), a far more abundant transcript with presumably more copies of primary transcripts. To further address this point, we repeated the chromatin fractionation experiment and analyzed for polyadenylated transcripts in each fraction. The additional requirement of polyadenylation should presumably bias against nascent transcripts, and select for processed, full-length transcripts. Polyadenylated Lethe RNA is still preferentially associated with chromatin compared to two control mRNAs (revised Figure 2—figure supplement 1), indicating that full length Lethe is associated with chromatin.

Nonetheless, the current data do not distinguish whether the functional pool of Lethe is on chromatin, in nucleoplasm, or both. When Lethe is fused to a SV40 polyadenylation signal (which will efficiently cause primary transcript processing and polyadenylation), Lethe expression inhibited NF-κB -dependent gene expression in a dose-dependent manner (Figure 4), suggesting that chromatin tethering as a primary transcript is not strictly required. We have added these considerations to the Discussion.

*4) The conclusion that Lethe is a “unique” pseudogene lncRNA that is induced by both inflammatory and anti-inflammatory therapeutic is an over-interpretation of the results presented*.

We agree with the assessment and have removed the word “unique”. Certainly other pseudogene lncRNAs may be proven to have similar functions in the future.

**References**

Hoffmann A, Levchenko A, et al. (2002). The IkappaB-NF-kappaB signaling module: temporal control and selective gene activation. *Science* 298(5596): 1241-1245. doi:10.1126/science.1071914 298/5596/1241.

Hung T, Wang Y, et al. (2011). Extensive and coordinated transcription of noncoding RNAs within cell-cycle promoters. *Nat Genet* 43(7): 621-629. doi:10.1038/ng.848.

Kawahara TL, Michishita E, et al. (2009). SIRT6 links histone H3 lysine 9 deacetylation to NF-kappaB-dependent gene expression and organismal life span. *Cell* 136(1): 62-74.

Kawahara TL, Rapicavoli NA, et al. (2011). Dynamic chromatin localization of sirt6 shapes stress- and aging-related transcriptional networks. *PLoS Genet* 7(6): e1002153. doi:10.1371/journal.pgen.1002153.

Kong L, Zhang Y, et al. (2007). CPC: assess the protein-coding potential of transcripts using sequence features and support vector machine. *Nucleic Acids Res* 35: W345-349. doi:10.1093/nar/gkm391.

Kretz M, Siprashvili Z, et al. (2013). Control of somatic tissue differentiation by the long non-coding RNA TINCR. *Nature* 493(7431): 231-235. doi:10.1038/nature11661.

Li J and Tibshirani R (2011). Finding consistent patterns: A nonparametric approach for identifying differential expression in RNA-Seq data. *Stat Methods Med Res*. doi:0962280211428386.